# Emergence of Shape Bias in Convolutional Neural Networks through Activation Sparsity

**Tianqin Li**
Carnegie Mellon University
tianqinl@cs.cmu.edu

**Ziqi Wen**
Carnegie Mellon University
ziqiwen@cs.cmu.edu

**Yangfan Li**
Nortwestern University
yangfanli2024@u.northwestern.edu

**Tai Sing Lee**
Carnegie Mellon University
tai@cnbc.cmu.edu

## Abstract

Current deep-learning models for object recognition are known to be heavily biased toward texture. In contrast, human visual systems are known to be biased toward shape and structure. What could be the design principles in human visual systems that led to this difference? How could we introduce more shape bias into the deep learning models? In this paper, we report that sparse coding, a ubiquitous principle in the brain, can in itself introduce shape bias into the network. We found that enforcing the sparse coding constraint using a non-differential Top-K operation can lead to the emergence of structural encoding in neurons in convolutional neural networks, resulting in a smooth decomposition of objects into parts and subparts and endowing the networks with shape bias. We demonstrated this emergence of shape bias and its functional benefits for different network structures with various datasets. For object recognition convolutional neural networks, the shape bias leads to greater robustness against style and pattern change distraction. For the image synthesis generative adversary networks, the emerged shape bias leads to more coherent and decomposable structures in the synthesized images. Ablation studies suggest that sparse codes tend to encode structures, whereas the more distributed codes tend to favor texture. Our code is host at the github repository: https://topk-shape-bias.github.io/

## 1 Introduction

Sparse and efficient coding is a well-known design principle in the sensory systems of the brain [3, 31]. Recent neurophysiological findings based on calcium imaging found that neurons in the superficial layer of the macaque primary visual cortex (V1) exhibit even a higher degree of lifetime sparsity and population sparsity in their responses than previously expected. Only 4–6 out of roughly 1000 neurons would respond strongly to any given natural image [37]. Conversely, a neuron typically responded strongly to only 0.4% of the randomly selected natural scene images. This high degree of response sparsity is commensurated with the observation that many V1 neurons are strongly tuned to more complex local patterns in a global context rather than just oriented bars and gratings [36]. On the other hand, over 90% of these neurons did exhibit statistically significant orientation tuning, though mostly with much weaker responses. This finding is reminiscent of an earlier study that found similarly sparse encoding of multi-modal concepts in the hippocampus [32]. This leads to the hypothesis that neurons can potentially serve both as a super-sparse specialist code with their **strong responses**, encoding specific prototypes and concepts, and as a more distributed code, serving as the classical sparse basis functions for encoding images with much **weaker responses**. The specialist

37th Conference on Neural Information Processing Systems (NeurIPS 2023).

code is related to the idea of prototype code, the usefulness of which has been explored in deep learning for serving as memory priors [24] in image generation, for representing structural visual concepts [38, 39], or for constraining parsimonious networks [26] for object recognition.

In computer vision community, recent studies found that Convolutional Neural Networks (CNNs) trained for object recognition rely heavily on texture information [11]. This texture bias leads to misclassification when objects possess similar textures but different shapes [2]. In contrast, human visual systems exhibit a strong 'shape bias' in that we rely primarily on shape and structures over texture for object recognition and categorization [20]. For instance, a human observer would see a spherical object as a ball, regardless of its texture patterns or material make-up [34]. This poses an interesting question: What is the design feature in the human vision systems that lead to the shape bias in perception?

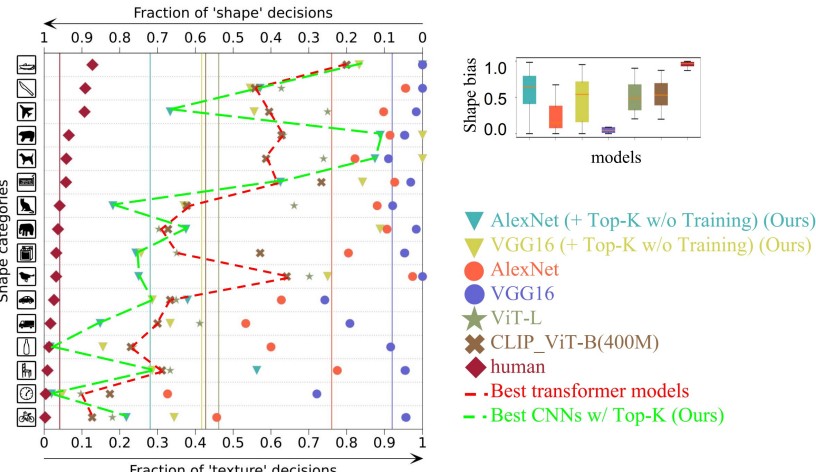

Figure 1: Shape bias of our sparse CNNs versus standard CNNs and SOTA transformer-based networks in comparison to the shape bias of human subjects, as evaluated on benchmark dataset [10] across 16 classes. The red dotted line shows the frontier of transformer-based networks with the best shape bias. The greed dotted line shows that sparse CNNs push the frontier of the shape bias boundary toward humans.

In this paper, we explore whether the constraint of the high degree of strong-response sparsity in biological neural networks can induce shape bias in neural networks. Sparsity, particularly in overcomplete representations, is known to encourage the formation of neurons encoding more specific patterns [30]. Here, we hypothesize that these learned specific patterns contain more shape and structure information, thus sparsifying the neuronal activation could induce shape bias in neuronal representation. To test this hypothesis, we impose a sparsity mechanism by keeping the Top K absolute response of neuronal activation at each channel in one or multiple layers of the network, and zeroing out the less significant activation with K is a sparsity parameter that we can adjust for systematic evaluation.

We found that this sparsity mechanism can indeed introduce more shape bias in the network. In fact, simply introducing the Top-K operation during inference in the pre-trained CNNs such as AlexNet [19] or VGG16 [33] can already push the frontier of the shape bias benchmark created by [10] (as shown in Figure 1). Additional training of these networks with the Top-K operation in place further enhances the shape bias in these object recognition networks. Furthermore, we found that the Top-K mechanism also improves the shape and structural bias in image synthesis networks. In the few-shot image synthesis task, we show that the Top-K operation can make objects in the synthesized images more distinct and coherent.

To understand why Top-K operation can induce these effects, we analyzed the information encoded in Top-K and non-Top-K responses using the texture synthesis paradigm and found that Top-K responses tend to encode structural parts, whereas non-Top-K responses contribute primarily to texture and color encoding, even in the higher layers of the networks. Our finding suggests that sparse coding is important not just for making neural representation more efficient and saving metabolic energy but

also for contributing to the explicit encoding of shape and structure information in neurons for image analysis and synthesis, which might allow the system to analyze and understand 3D scenes in a more structurally oriented part-based manner [4], making object recognition more robust.

## 2   Related Works

**Shape Bias v.s. Texture Bias**   There has been considerable debate over the intrinsic biases of Convolutional Neural Networks (CNNs). [11] conducted a pivotal study demonstrating that these models tend to rely heavily on texture information for object recognition, leading to misclassifications when objects have similar textures but distinct shapes. In addition, it has also been shown that using the texture information alone are sufficient to achieve object classification [6]. This texture bias contrasts markedly with human visual perception, which exhibits a strong preference for shape over texture – a phenomenon known as 'shape bias'[20]. Humans tend to categorize and recognize objects primarily based on their shape, a factor that remains consistent across various viewing conditions and despite changes in texture [34]. These studies collectively form the foundation upon which our work builds, as we aim to bridge the gap in shape bias between computer vision systems and human visual systems.

**Improving Shape Bias of Vision Models**   Following the identification of texture bias in CNNs by [11], numerous studies sought to improve models' shape bias for better generalization. Training methods have been developed to make models more shape-biased, improving out-of-distribution generalization. Some approaches, like [11], involved training with stylized images to disconnect texture information from the class label. Such approach posed computational challenges and didn't scale well. Others, like [15], used human-like data augmentation to mitigate the problem, while [23] proposed shape-guided augmentation, generating different styles on different sides of an image's boundary. However, these techniques all rely on data augmentation. Our focus is on architectural improvements for shape bias, similar to [1] which created a texture-biased model by reducing the CNN model's receptive field size. We propose using sparsity operations to enhance shape bias of CNNs. Furthermore, [7] proposes to scale up the transformer model into 22 billion parameters and show a near human shape bias evaluation results. We, on the other hand, are not comparing with their network since we focus on CNNs which requires less computation and doesn't require huge data to learn. We demonstrate in the supplementary that the same sparsity constraint could also be beneficial to the ViT family as well, hinting the generalizability of our findings.

**Robustness In Deep Learning**   Robustness in deep learning literature typically refers to robustness against the adversarial attack suggested by [35] which showed that minuscule perturbations to images, imperceptible to the human eye, can drastically alter a deep learning model's predictions. Subsequent research [14, 21] corroborated with these findings, showing that deep neural networks (DNNs) are vulnerable to both artificially-induced adversarial attacks and naturally occurring, non-adversarial corruptions. However, the robustness we are mentioning in this paper is about the robustness against confusing textures that are misaligned with the correct object class, as illustrated by the cue-conflict datasets provided by [11]. Although sparsity has been shown to be effective against the adversarial attack [25], explicit usage of Top-K in shape bias has not been explored.

## 3   Method

### 3.1   Spatial Top-K Operation in CNN

**Sparse Top-K Layer**   We implement the sparse coding principle by applying a Top-K operation which keeps the most significant K responses in each channel across all spatial locations in a particular layer. Specifically, for a internal representation tensor $X \in R^{c \times h \times w}$, Top-K layer produces $\texttt{X\_Top\_K} := \texttt{Top\_K(X, K)}$, where the $\texttt{X\_Top\_K}$ is defined as:

$$\texttt{X\_Top\_K}[\texttt{i},\texttt{j},\texttt{k}] := \begin{cases} \texttt{X}[\texttt{i},\texttt{j},\texttt{k}], & \text{if } \texttt{abs}(\texttt{X}[\texttt{i},\texttt{j},\texttt{k}]) \geq \texttt{Rank}(\texttt{abs}(\texttt{X}[\texttt{i},:,:]))[K] \\ 0, & \text{otherwise} \end{cases} \tag{1}$$

Equation 1 specifies how each entry of a feature tensor $X \in R^{c \times h \times w}$ would be transformed inside the Top-K layer. The zero-out operation in Equation 1 suggests that the gradient w.r.t. any non Top-K

value, as well as the gradients that chain with it in the previous layers will become zero. However, our analysis later suggests that the network can still learn and get optimized, leading to improved dynamic spatial Top-K selection.

**Sparse Top-K With Mean Replacement**    To determine the relative importance of the Top-K values or the Top K positions in the Top-K operation, we create an alternative scheme in which all the Top-K responses in a channel are replaced by the mean of the Top-K responses in the channel `Top_K_Mean_Rpl` as defined below:

$$X_{\texttt{Top\_K\_Mean\_Rpl}}[i,j,k] := \begin{cases} \frac{1}{n}\sum_{j,k\in\{\text{Top-K}\}_n} X[i,j,k], & \text{if } \texttt{abs}(X[i,j,k]) \geq \texttt{Rank}(\texttt{abs}(X[i,:,:]))[K] \\ 0, & \text{otherwise} \end{cases}$$

This `Top_K_Mean_Rpl` operation reduces the communication rate between layers by 1000 times. We study the impact of this operation on object performance in an object recognition network (See section 4.3 for the results) and to determine which type of information (values versus positions) is essential for inducing the shape bias.

## 3.2    Visualizing the Top-K code using Texture Synthesis

We used a texture synthesis approach [9] with ablation of Top-K responses to explore the information contained in the Top-K responses in a particular layer using the following method.

Suppose a program $F(\cdot)$ denotes a pre-trained VGG16 network with a number of parameters N [33] and $TS: R^{h\times w\times 3} \times N \to R^{h\times w\times 3}$ denotes the texture synthesis program from [9] where an input image $I$ is iteratively optimized by gradient descent to best match the target image $T$'s internal activation when passing through $F(T)$. We detail the operations inside $TS$ below. Denote the internal representation at each layer $i$ of the VGG16 network when passing an input image $I$ through $F(\cdot)$ as $X_i(I)$, and suppose there exist $L$ layers in the VGG16 network. We update the image $I$ as follows:

$$I \leftarrow I - lr * \left(\frac{\partial}{\partial I}\sum_i^L [Gr(X_i(I)) - Gr(X_i(T))]\right)$$

,where $Gr(\cdot): R^{h\times w\times c} \to R^{c\times c}$ denotes the function that computes gram matrix of a feature tensor, i.e. $Gr(X_i(I)) = X_i(I)^T X_i(I)$. We adopt LBFGS [28] with initial learning rate 1.0 and 100 optimization steps in all our experiments with the texture synthesis program.

Utilizing the above texture synthesis program $TS(\cdot, \texttt{VGG16})$, we can obtain the synthesis results in $S_{\text{w/o Top-K}}$ by manipulating the internal representation of VGG16 such that we only use the non-Top-K responses to compute the Gram matrix when forming the Gram matrix optimization objectives. This effectively computes a synthesis such that it only matches with the internal non-Top-K neural response. For a given target image $T$, this leads to $S_{\text{w/o Top-K}}$:

$$S_{\text{w/o Top-K}} = TS(T, \texttt{ZeroOutInternalTopK(VGG16)})$$

, which would show the information encoded by non-Top-K responses. Next, we include the Top-K firing neurons when computing the Gram matrix to get $S_{\text{w/ Top-K}}$:

$$S_{\text{w/ Top-K}} = TS(T, \texttt{IdentifyFunction(VGG16)})$$

. Comparing these two results will allow us assess the information contained in the Top-K responses.

## 3.3    Visualizing the Top-K neurons via Reconstruction

Similar to section 3.2, we provide further visualization of the information Top-K neurons encode by iteratively optimizing an image to match the internal Top-K activation directly. Mathematically, we redefine our optimization objective in section 3.2:

$$I \leftarrow I - lr * \left(\frac{\partial}{\partial I}\sum_i^L [X_i(I) - \texttt{Mask}_i * X_i(T)]\right)$$

, where $\texttt{Mask}_i$ a controllable mask for each layer $i$. There are three types of mask we used in the experiments: {`Top-K_Mask`, `non_Top-K_Mask`, `Identity_Mask`}. `Top-K_Mask` selects only the Top-K fired neurons while keeps the rest of the neurons zero, whereas `non_Top-K_Mask` only selects the opposite of the `Top-K_Mask` and `Identity_Mask` preserves all neurons. By comparing these three settings, one can easily tell the functional difference between Top-K and non Top-K fired neurons (See results in Figure 3).

### 3.4 Shape Bias Benchmark

To demonstrate our proposal that the Top-K responses are encoding the structural and shape information, we silence the non-Top-K responses during inference when using pre-trained CNNs. To test the networks' shape bias, we directly integrate our code into the benchmark provided by [10]. The benchmark contains a cue-conflict test which we use to evaluate the Top-K operation. The benchmark also includes multiple widely adopted models with pre-trained checkpoints and human psychological evaluations on the same cue-conflict testing images.

## 4 Results

### 4.1 Top-K Neurons Encode Structural Information

To test the hypothesis that the shape information is mostly encoded among the Top-K significant responses, whereas the non-Top-K responses are encoding primarily textures, we used the method described in Section 3.2 and compared the texture images synthesized with and without the Top-K responses for the computation of the Gram matrix. Figure 2 compares the TS output obtained by matching the early layers and the higher layers of the VGG16 network in the two conditions. One can observe that ablation of the Top-K responses eliminated much of the structural information, resulting in more texture images. We conclude from this experiment that (1) Top-K responses are encoding structural parts information; (2) Non Top-K responses are primarily encoding texture information.

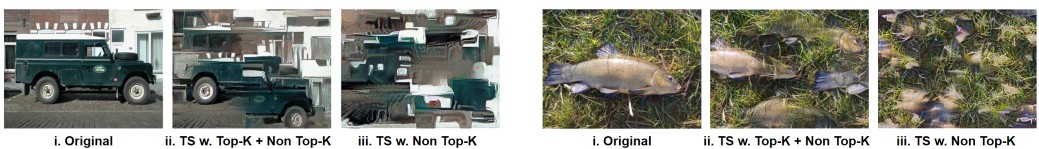

Figure 2: Texture Synthesis (TS) using [9]. i. shows the original image, ii. shows the TS results of $S_{\text{w/ Top-K}}$ with both Top-K and Non Top-K activation intact, iii. shows the TS results $S_{\text{w/o Top-K}}$ with Top-K activation deleted before performing TS.

To provide further insights about the different information Top-K and non Top-K neurons are encoding, we show another qualitative demonstration in Figure 3 where we optimize images that would excite the Top-K neurons alone and the non Top-K neurons alone respectively (See full description in Section 3.3).

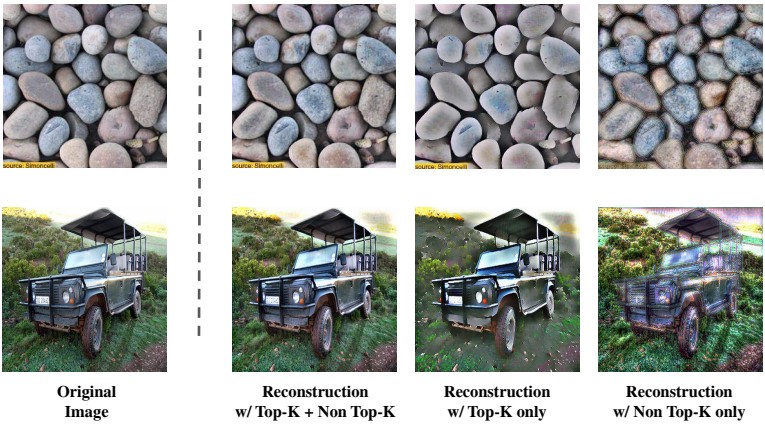

Figure 3: Visualizing Top-K and non Top-K neurons through optimizing input images to match their activation.

From Figure 3, it is clear that optimizing images to match the Top-K fired neurons yields high level scene structures with details abstracted away while optimization efforts to match the non Top-K fired neurons produce low level local textures of the target images. Together, we provide evidence to

support our hypothesis that it is the strong firing neurons in the convolutional neural networks that provide structural information while the textures are encoded among the weakly activated neurons. Next, we demonstrate that this phenomenon will result in improved shape bias in both analysis and synthesis tasks.

## 4.2 Top-K Responses already have Shape Bias without Training

We test the Top-K activated CNNs with different degrees of sparsity on the shape bias Benchmark proposed from [10]. This benchmark evaluates the shape bias by using a texture-shape cue conflict dataset where the texture of an image is replaced with the texture from other classes of images. It defines the shape and texture bias in the following ways:

$$\text{shape bias} = \frac{\texttt{\# of correct shape recognitions}}{\texttt{\# of correct recognitions}}$$

$$\text{texture bias} = \frac{\texttt{\# of correct texture recognitions}}{\texttt{\# of correct recognitions}}$$

It has been shown in previous work [10, 11] that CNNs perform poorly on shape-based decision tests whereas human subjects can make successful shape-based classification on nearly all the evaluated cases. This results in CNN models having relatively low shape bias scores while humans have close to 1 shape bias score. Interestingly, it has been observed that Vision Transformers (ViT) model family has attained significant improvement in shape bias [10].

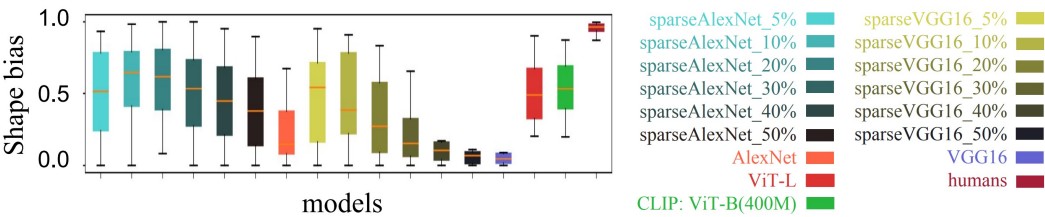

Figure 4: Overall shape bias of sparse CNNs, CNNs, Transformer and humans

Adding the Top-K operation to a simple pretrained such as AlexNet or VGG16 alone already can already induced a significant increase in shape bias, as shown in Fig.4. With the sparsity knob K equal to 10% and 20%, the Top-K operation alone appears to achieve as much or more shape bias as the state-of-the-art Vision Transformer models in the cue-conflict dataset, leading further support to the hypothesis that Top-K sparsity can lead to shape bias.

We plot the best of the Top-K sparsified AlexNet and VGG16 for each evaluation of 16 object classes in Fig.5. We can observe that sparsity constraint improve shape-biased decision-making for most of the object classes, bringing the performance of the pre-trained model closer to human performance. With the proper settings of sparsity, certain classes (e.g. the bottle and the clock category) could attain human level performance in shape-biased scores.

However, we should note that the confidence interval is quite large, indicating that the network performs differently across the different classes. A closer look at shape bias for each class is shown in Figure 5.

## 4.3 Top-K training induces Shape Bias in Recognition Networks

To evaluate the shape bias can be enhanced by training with Top-K operation, we trained ResNet-18 [13] on different subsets of ImageNet dataset [8]. Each subset contains randomly selected 10 original categories from ImageNet,for all the training and evaluation. Every experiment is run three times to obtain an error bar. During the evaluation, we employ AdaIn style-transfer using programs adopted from [18] to transform the evaluation images into a texture-ablated form as shown in Figure 6. The original texture of the image is replaced by styles of non-related images using style transform.

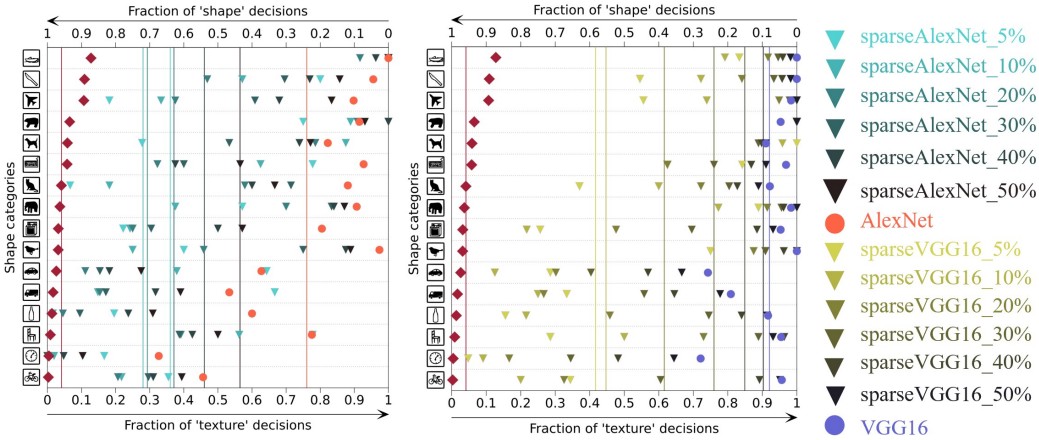

Figure 5: The classification result on the Shape Bias Benchmark proposed from[10]. This plot shows the shape bias of sparse CNNs, CNNs and humans on different class in texture-shape cue conflict dataset. It also show the shape bias of different sparsity degree. e.g. 5% means that only top 5% activation value would be passed to the next layer. Vertical lines means the average value.

This allows us to evaluate how much a trained model is biased toward the original texture instead of the underlying shape.

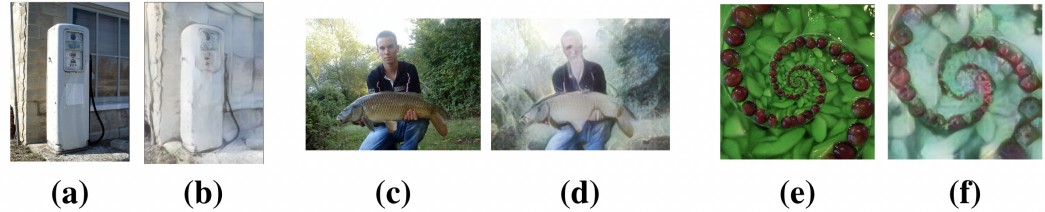

|  (a)  |  (b)  |  (c)  |  (d)  |  (e)  |  (f)  |

Figure 6: Evaluating shape bias of the network with stylized ImageNet subsets. Three pairs of images are presented sampled from the our evaluation datasets. Specifically, we transfer (a) → (b), (c) → (d) and (e) → (f) by AdaIN [18] and keep the original class labels. During the evaluation, the transferred images are presented instead of the original test image to measure the network's texture bias sensitativity.

In this experiment, we train classification networks with two non-overlapping subsets of the ImageNet, namely IN-$S_1$ and IN-$S_2$. We select the categories that are visually distinctive in their shapes. The details about the datasets can be found in the Supplementary Information. We trained ResNet-18 models over the selected IN-$S_1$ and IN-$S_2$ dataset with the standard Stochastic Gradient Descent (SGD, batch size 32) and a cosine annealing learning rate decay scheduling protocol with lr starting from 0.1. The same optimization is applied to ResNet-18 models with a 20% spatial Top-K layer added after the second bottleneck block of the ResNet-18. All models are then evaluated on the styleized version of IN-$S_1$ and IN-$S_2$ evaluation dataset after trained with 50 epochs.

| Top-1 Acc. (%) | IN-$S_1$ (↑) | Stylized-IN-$S_1$ (↑) | IN-$S_2$ (↑) | Stylized-IN-$S_2$ (↑) |
|---|---|---|---|---|
| ResNet-18 [13] | 87.8 ± 0.5 | 49.3 ± 1.5 | 81.3 ± 1.7 | 52.4 ± 2.2 |
| ResNet-18 w. Top-K during training | **89.4** ± 0.6 | **55.4** ± 0.8 | 83.4 ± 0.9 | **59.7** ± 0.6 |
| ResNet-18 w. Top_K_Mean_Rpl during training | 84.9 ± 0.3 | 56.8 ± 1.7 | 75.5 ± 2.5 | 53.1 ± 1.0 |

Table 1: Evaluation for models trained on IN-$S_1$ and IN-$S_2$ datasets, each of which consists 10 classes of all train/val data from ImageNet-1k dataset [8].

Table 1, shows that (i) the classification accuracy on the original evaluation dataset doesn't drop: mean top-1 accuracy of 87.8 (baseline) v.s. 89.4 (w. Top-K) on IN-$S_1$ and 81.3 (baseline) v.s. 83.4 (w.

Top-K) on IN-$S_2$ respectively even when we push the sparsification to K= 20%; (ii) The shape bias improves significantly: mean top-1 accuracy of 55.4 (w. Top-K) v.s. 49.3 (baseline) on Stylized-IN-$S_1$ and 59.7 (w. Top-K) v.s 52.4 (baseline) on Stylized-IN-$S_2$ respectively. This supports our conjecture that the sparse code could introduce more shape bias, in comparison to the dense representation, during learning.

To further investigate why Top-K might induce shape bias, we evaluate whether the values of the Top-K responses matter by compressing the information in each channel to the mean responses of the Top-K responses of that channel at the Top-K positions. This reduces the information of each channel to only a binary mask indicating the Top-K responding locations and a single float number that relays the channel's weighted contribution to downstream layers, effective compressing the communication rate by 3 orders of magnitude (See Section 3.1 for a detailed description of the Top_K_Mean_Rpl).

Despite the enormous amount of data compression by replacing the Top-K values with the mean, the network can still maintain the shape bias comparable to the normal ResNet-18 baseline (as indicated by the improved or on-par performance on the Stylized-IN-$S_1$/$S_2$ between ResNet-18 and ResNet-18 w. Mean Top-K in Table 1). This suggests that the spatial map of the Top-K activation is more important than the precise values of the Top-K responses. This suggests a significant amount of the object shapes features are actually encoded in the occupancy map of significant neural activities, i.e. the binary mask of the Top-K.

### 4.4 Towards Shape Biased Few Shot Image Synthesis

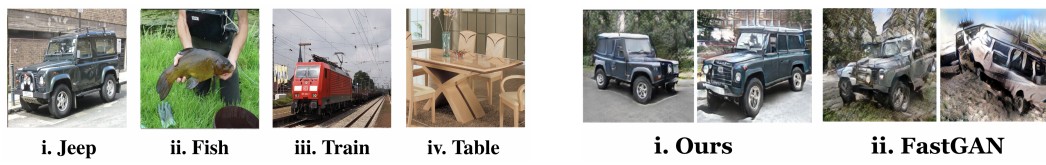

(a) Image samples from the four training categories.    (b) Qualitative Comparison on Jeep-100 Synthesis.

Figure 7: Few shot image synthesis datasets and qualitative comparison results between our methods and FastGAN [27].

Humans are great few-shot learners, i.e. learning from a few examples. This ability might be related to our cognitive ability to learn a generative model of the world that allows us to reason and imagine. Recurrent feedback in the hierarchical visual system has been hypothesized to implement such a generative model. We investigate whether the Top-K operation also induces shape bias in the synthesis network for the few-shot image synthesis task. We hypothesize that shape bias can benefit few-shot image synthesis by emphasizing on structural and shape information. Figure 7b shows that state-of-the-art few-shot synthesis program (FastGAN [27]) suffers severely from texture bias. We found that introducing Top-K operation in the fourth layer (the 32 x 32 layer) in FastGAN, significant improvement in the synthesis results can be obtained on datasets selected from ImageNet [8] (100 samples each from four diverse classes, synthesizing each class independently, see Supplementary for details) as shown in Table 7. Images from ImageNet possess rich structural complexity and population diversity. To be considered to be a good synthesis, generated samples would have to achieve strong global shape coherence in order to have good matching scores with the real images. A better quantitative evaluation result would suggest the emergence of a stronger shape or structural bias in the learned representation. Samples of the four classes are shown in Figure 7a.

To assess the image synthesis quality, we randomly sample 3000 latent noise vectors and pass them through the trained generators. The generated images are then compared with the original training images in the inception-v3 encoder space by Fréchet Inception Distance (FID [16]) and Kernel Inception Distance (KID [5]) scores and documented in Table 2. Each setting is run 3 times to produce an error bar.

First, the synthesis quality measurements for adding the Top-K operation to the FastGAN network show a consistent improvement in terms of FID and KID scores in Table 2. Figure 7b shows that the Top-K operation leads the generation of objects (e.g. the jeep) that are more structurally coherent and distinct, compared to the original FastGAN. Specifically, we observe a 21.1% improvement on FID scores and 50.8% improvement on KID scores for the Jeep-100 class when K= 5% sparsity was

imposed during training (i.e. only keep 5% neurons active). Similarly, when 5% sparsity is imposed on Fish-100 and Train-100 datasets the FID is increased by 17.3% and 12% respectively and the KID performance is boosted by 48.5% and 33.4%. Lastly, we test the in-door table class which contains complex objects with many inter-connected parts and subparts. A K=15% sparsity leads to a gain of 9.3 % and 22.3% in FID and KID respectively for synthesizing similar in-door tables. Overall, our experiments show that introducing the Top-K operation in a challenging few-shot synthesis task can significantly improve the network's generated results on a diverse set of complicated natural objects.

| Method | Jeep-100 | | Fish-100 | | Train-100 | | Table-100 | |
|---|---|---|---|---|---|---|---|---|
| | FID ↓ | KID* ↓ | FID ↓ | KID* ↓ | FID ↓ | KID* ↓ | FID ↓ | KID* ↓ |
| FastGAN [27] | $49.0 \pm 1.4$ | $12.0 \pm 1.9$ | $46.2 \pm 1.8$ | $13.4 \pm 0.6$ | $46.1 \pm 1.8$ | $11.2 \pm 0.8$ | $67.2 \pm 0.1$ | $19.7 \pm 0.3$ |
| FastGAN w. Top-K (ours) | $\mathbf{38.7 \pm 0.5}$ | $\mathbf{5.9 \pm 0.7}$ | $\mathbf{38.2 \pm 1.4}$ | $\mathbf{6.9 \pm 0.7}$ | $\mathbf{40.2 \pm 0.8}$ | $\mathbf{7.4 \pm 0.2}$ | $\mathbf{60.9 \pm 0.3}$ | $15.3 \pm 0.2$ |

Table 2: Few-shot Image Synthesis results measured in FID [17] and KID [5] Note that KID * denotes KID scaled by a factor of $10^3$ to demonstrate the difference.

## 4.5 Parts Learning in Sparse Top-K

Finally, we study the training dynamics of the Top-K layer's internal representation. In Figure 8, we can make the following observations. (1) By applying sparse constraint using Top-K, each channel is effectively performing a binary spatial segmentation, i.e. the image spatial dimension is separated into either Top-K region or non Top-K territory. (2) Although there is no explicit constraint to force the Top-K neurons to group together, the Top-K responses tend to become connected, forming object parts and subparts, as training evolves.

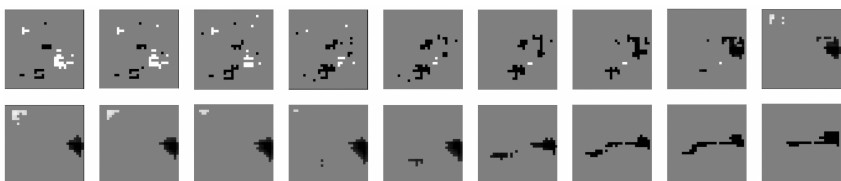

Figure 8: Even though Top-K Operation is not fully differentiable, the network is able to relocate the spatial activation mass smoothly towards a connected meaningful parts which eventually leads to component learning as shown in Figure 9

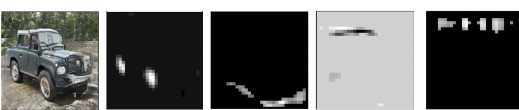

Figure 9: Synthesis network internal topk layers reveals semantic decomposition of parts and subparts.

We believe the development of this continuous map when training with Top-K operation might be due to two factors (1) CNN activations are locally smooth, i.e. two adjacent pixels in layer $L_n$ are linked by the amount of overlap between their corresponding input patches in $L_{n-1}$, (2) Top-K increase the responsibility of each individual neuron to the loss function. When neurons i and j are being selected as Top-K, their responses are likely similar to each other. However, if their corresponding spatial location in the output has different semantic meanings, they will receive diverse gradients which will be then amplified by the increased individual responsibility. The diverging gradients would lead to the value difference of two neuron i and j's gradients, resulting in one of them leaving the Top-K group while only the semantic similar ones remaining the Top-K sets. This might suggest a principle we call *neurons fire together, optimize together* during CNN Top-K training, which could lead to the observed emerging of semantic parts and subparts. This localist code could further connect to recognition by component theory [4, 22, 12] which constranints the representation by semantic parts segmentation, improving the network's focus on object components boundary, hence improving overall shape bias. In functional analysis of the brain, [29] also show that a local smoothness constraint could lead to

the topological organization of the neurons, hinting that the hypothesized factors above could have a neuroscientific grounding.

**Top-K Sparsity Hyperparameter**    With the understanding from Section 4.5, we want to reiterate the importance of the sparsity hyperparameter we used. The amount of the sparsity can be directly translated to "size of the parts". Thus, depending on the image type, the composition of the scene, the network architecture, and the layers in which the Top-K sparsity operation is applied on, the results could be drastically different. We refers to the supplementary for more detailed ablation study.

## 5    Conclusion

In this study, we discovered that an operation inspired by a well-known neuroscience design motif of sparse coding can induce shape bias in neural representation. We demonstrated this in object recognition networks and in few-shot image synthesis networks. We found that simply adding the Top-K sparsity operation can induce shape bias in pre-trained convolutional neural networks and that training of the CNNs and GAN with the simple Top-K operation can increase the shape bias further toward human performance, which makes object recognition more robust against texture variations and makes image synthesis generating structurally more coherent and distinct objects. Using texture synthesis, we are able to demonstrate that Top-K responses carry more structural information, while the non Top-K responses carry more texture information. The observation that sparse coding operation can induce shape bias in deep learning networks suggests sparsity might also contribute to shape bias in human visual systems.

## 6    Ethics Statement

This study investigates whether the sparse coding motif in neuroscience can induce shape bias in deep learning networks. The positive results suggest that sparsity might also contribute to shape bias in the human visual systems, thus providing insights to our understanding of the brain. While deep learning can advance science and technology, it comes with inherent risks to society. We acknowledge the importance of ethical study in all works related to deep learning. A better understanding of deep learning and the brain however is also crucial for combating the misuse of deep learning by bad actors in this technological arm race.

## 7    Acknowledgement

This work was supported by an NSF grant CISE RI 1816568 awarded to Tai Sing Lee. This work is also partially supported by the graduate student fellowship from CMU Computer Science Department.

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
