# Supplementary for Emergence of Shape Bias in Convolutional Neural Networks through Activation Sparsity

## 1 Further Results of the impact of sparsity on Shape Bias Benchmark

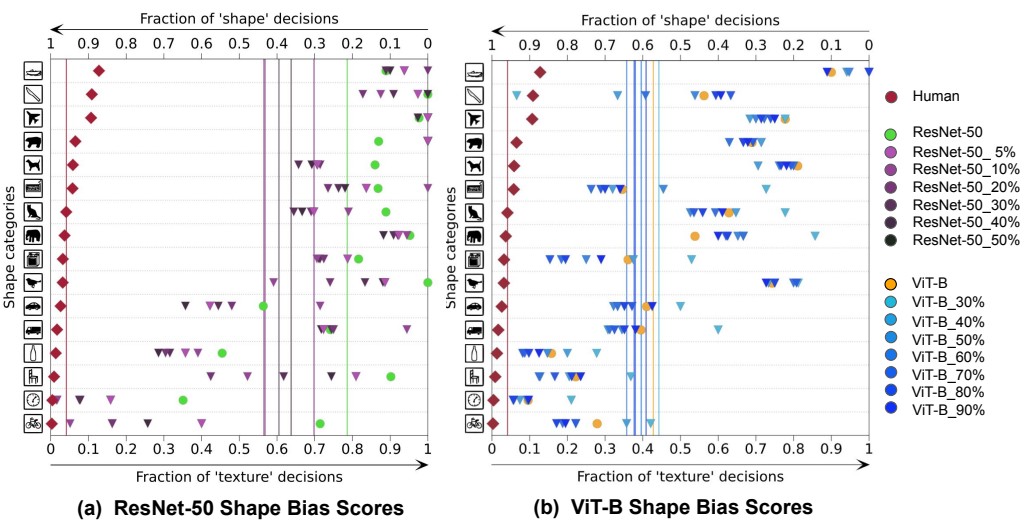

(a) ResNet-50 Shape Bias Scores        (b) ViT-B Shape Bias Scores

Figure 1: Additional Inference with Top-K results on ResNet-50 pretrained on ImageNet-1k and ViT-B pretrained on ImageNet-21k. We utilize the sparsity operation proposed in Section 3.1 for ResNet-50. For ViT, we also apply the spatial Top-K operation as described in the general response. We can observe an increase in both ResNet-50 and ViT-B architectures, furthering closing the gap between human and existing models.

We generalize section 4.2 in the main text to ResNet-50 and ViT-B architectures (Figure 1). The ResNet-50's sparsity definition is the same as AlexNet and VGG. For ViT-B, we reshape the intermediate activation response from [n, h * w, d] to [n, d, h * w] and apply the Top-K selection over dimension 2 before the activation is passed through the multiple head attention (Note that h and w is the height and weight of the latent tensor after reshape it to 2d, for ViT-B with patch size 16 on the 224x224 images, h=w=14, n denotes the batch size). Please refer to our updated code for precise implementation. We observe that the similar shape bias gain from applying Top-K operation to ResNet-50 and ViT-B although ViT-B needs denser activation compared to the ConvNets.

Interestingly, our additional experiments in the Figure 1 indicates that ViT architectures seem to require more dense activation, as we can observe that a moderate to high density (keep 50% - 80% of the original ranking neurons) yield the best performance for most of the classes (best mean shape score is achieved by keeping 60% of the original activation), however, the optimal sparsity rates in AlexNet and VGG are around 5 - 10%.

37th Conference on Neural Information Processing Systems (NeurIPS 2023).

## 2 Top-K layers configurartion

We apply the Sparsity layer in a subset of the network. It is based on the intuition that the brain utilizes sparsity for long range communication but can allow local dense computation. We divide the networks into chunks where within each chunk the neuron's activities are allowed to be dense (keep original) but the communication across different chunks is set to be sparse. To be specific, in the VGG Network, we divides the network into 5 chucks and apply the Top-K layer after each MaxPooling layer. Specifically, across a total of 31 feature layers in VGG, we apply the Top-K layer after the 5th, 10th, 17th, 23th, 31th original layers. In AlexNet, we also apply the Top-K layer after the MaxPooling except adding 2 additional Top-K layers at the 8th and 10th layer to make the sparse layer more evenly distributed. In the ResNet-18 training (Main Section 4.3), we only apply the Top-K layer after the second layer to demonstrate the concept of sparse specialist code. We also apply the Top-K layer only in the 4th layer of the GAN generator as we observe empirically that it induces the best performance across other settings (such as applying multiple Top-K layers or the less severe sparsity).

## 3 More qualitative results of image synthesis

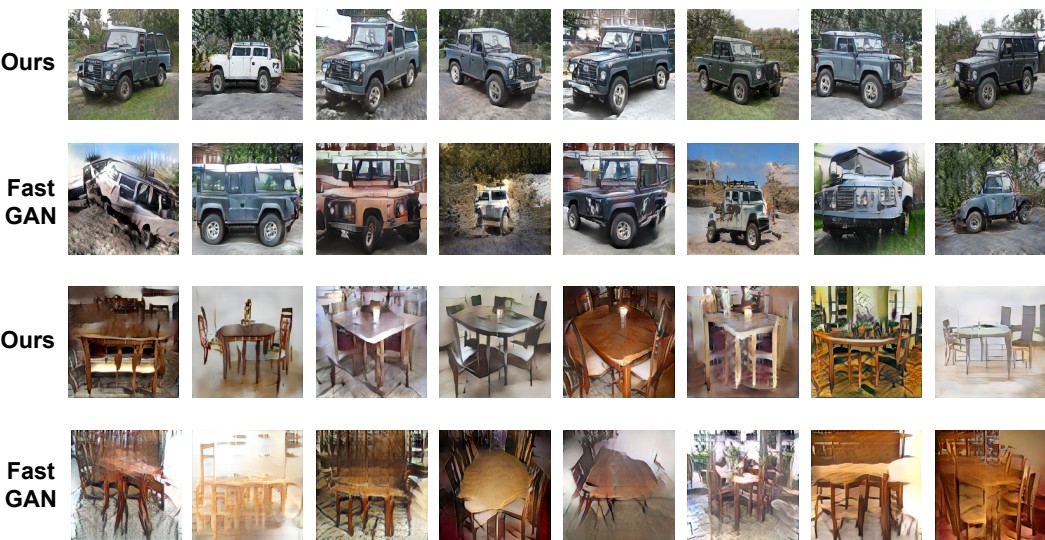

Figure 2: Qualitative comparison between our method and FastGAN [8] baseline.

We show more samples from our trained generator and the FastGAN baseline [8] and show a few of them in Figure 2. As we can see that with the help of sparsity, the generated samples are more structurally coherent comparing to the one without sparsity. We hypothesize that the improved synthesis quality comes from the increased focus on the shape bias of the generator CNNs. By excluding the activation of neurons outside the Top-K range, we effectively avoid the negative impact of under-trained and poorly performing texture synthesis on the overall quality of synthesized examples. This visual examination further supports that few-shot image synthesis is a good application where a network with shape biased generator would be superior for achieving cohesive and appropriate synthesis quality.

## 4 The Impact of sparsity on Top-K image synthesis

From Section 4.5 of the main manuscript, we can see that an individual sparsified feature channel is encoding an object part. Thus, the optimal Top-K sparsity might depend on the complexity of the object and the amount of data, and how it can be best decomposed into individual parts, as shown in Figure 4. Insufficient K (too sparse) can force the breakup of a coherent part from one channel

into multiple subchannels. Excessive K can lead to rigidity and interference of parts into the same channels. Therefore, the optimal K might vary with the complexity of images in few-shot synthesis performance. For example, Jeep-100 reaches the best performance at 5% sparsity with the mean FID score of 38.7, whereas the best Table class synthesis (mean FID 60.9) is achieved at 15% sparsity. The image synthesis performance results at different sparsity levels are shown in the Table.

| Method | Jeep-100 | | Table-100 | |
| --- | --- | --- | --- | --- |
| | FID ↓ | KID* ↓ | FID ↓ | KID* ↓ |
| FastGAN [8] | $49.0 \pm 1.4$ | $12.0 \pm 1.9$ | $67.2 \pm 0.1$ | $19.7 \pm 0.3$ |
| FastGAN w. Top-K 2% (ours) | $42.2 \pm 1.5$ | $8.2 \pm 0.7$ | $86.6 \pm 0.4$ | $36.1 \pm 0.5$ |
| FastGAN w. Top-K 5% (ours) | $\mathbf{38.7} \pm 0.5$ | $\mathbf{5.9} \pm 0.7$ | $80.1 \pm 1.1$ | $34.5 \pm 0.9$ |
| FastGAN w. Top-K 7% (ours) | $44.5 \pm 0.8$ | $8.1 \pm 0.4$ | $62.9 \pm 0.6$ | $20.0 \pm 1.7$ |
| FastGAN w. Top-K 10% (ours) | $45.3 \pm 0.5$ | $9.0 \pm 1.8$ | $61.7 \pm 1.2$ | $16.1 \pm 1.0$ |
| FastGAN w. Top-K 15% (ours) | $43.2 \pm 1.2$ | $9.6 \pm 0.5$ | $\mathbf{60.9} \pm 0.3$ | $\mathbf{15.3} \pm 0.2$ |
| FastGAN w. Top-K 20% (ours) | $52.1 \pm 1.0$ | $20.6 \pm 1.3$ | $64.9 \pm 1.0$ | $17.5 \pm 0.9$ |
| FastGAN w. Top-K 50% (ours) | $47.3 \pm 0.9$ | $13.6 \pm 0.6$ | $66.3 \pm 0.7$ | $17.3 \pm 0.4$ |

Table 1: Few-shot Image Synthesis ablation study. Image synthesis quality is measured in FID [5] and KID [1] Note that KID * denotes KID scaled by a factor of $10^3$ to demonstrate the difference.

## 5   Further experiments on Top-K training

We further expand the experiments for 4.3 to demonstrate more details of the improvements on out-of-distribution test. We first test how the model behaves when we apply Top-K operation on Non-Top-K trained models. As the choice of hyper-parameter K in the proposed Top-K operation is critical to the model's performance, we apply the same Top-K parameter setting for (2) and (4). It can be observed that inference with Top-K with sub-optimal setting (2) would sometimes induce performance deficit but the models' performance will improve when applying the same Top-K sparsity during training. Second, we test the hypothesis that whether other soft sparsity constraints like L1 regularization would have the same effect. We introduce a L1 term ( $+ \lambda$ |NeuronActivation|) during training loss (we tried choice of $\lambda = \{10, 1, 0.1\}$ and found $\lambda = 1$ works the best). We present this form of sparsity results in (5). We can observe that (5) has similar performance as trained with Top-K layers (4), suggesting that L1 regularization induced sparsity could also help for improve the out-of-domain generalization of the model.

| Top-1 Acc. (%) | IN-$S_1$ (↑) | Stylized-IN-$S_1$ (↑) | IN-$S_2$(↑) | Stylized-IN-$S_2$ (↑) |
| --- | --- | --- | --- | --- |
| (1) ResNet-18 | $87.8 \pm 0.5$ | $49.3 \pm 1.5$ | $81.3 \pm 1.7$ | $52.4 \pm 2.2$ |
| (2) ResNet-18 w. Top-K inference (parameter setting 1) | $43.8 \pm 0.8$ | $23.4 \pm 1.0$ | $58.0 \pm 1.6$ | $42.3 \pm 3.3$ |
| (3) ResNet-18 w. Best Top-K inference (parameter setting 2) | $87.4 \pm 0.4$ | $53.2 \pm 0.5$ | $81.2 \pm 1.2$ | $56.8 \pm 2.4$ |
| (4) ResNet-18 w. Top-K training (parameter setting 1) | $\mathbf{89.4} \pm 0.6$ | $\mathbf{55.4} \pm 0.8$ | $83.4 \pm 0.9$ | $\mathbf{59.7} \pm 0.6$ |
| (5) ResNet-18 w. L1 sparse activation regularization | $89.8 \pm 0.3$ | $55.5 \pm 0.5$ | $82.6 \pm 0.5$ | $54.4 \pm 1.1$ |
| (6) ResNet-18 w. Top_K_Mean_Rpl during training | $84.9 \pm 0.3$ | $56.8 \pm 1.7$ | $75.5 \pm 2.5$ | $53.1 \pm 1.0$ |

Table 2: The effect of activation sparsity to both in-domain generalization and out-of-domain generalization. (1) represents the baseline model of ResNet-18; (2) utilize the same sparse K parameter setting as (4) but (4) trained with Top-K and (2) doesn't; (3) uses an optimal Top-K sparsity for inference; (5) L1 activation constraint during training is applied.

## 6   Details of experimental setup

### 6.1   Texture synthesis experiment

**Dataset**   We utilize two images from the ImageNet dataset ( `n03594945_3233.jpg` and `n01440764_1764.jpg`) to run the modified texture synthesis program we developed in the Section 3.2. The data can be found from the ImageNet ILSVRC 2012 challenge [2]. The data is available for free to researchers for non-commercial use.

**Network Architecture and Optimization Parameters**    In this section, we modifed VGG16 [9] to perform texture synthesis. In particular, we install Top-K control layers in `relu1_1`, `pool1`, `pool2`, `pool3` and `pool4` layers and assigned an equal weight of 1e+09 when computing the objectives for optimization. For all Top-K and non Top-K selection, we employ a sparsity of 5% for both images. LBFGS optimization is used with learning rate of 1.0 and optimization steps of 100. We observe that all optimizations eventually converge as the loss values reach plateau at the end of the process. Our model is reproducible via the github repo provided at `https://github.com/Crazy-Jack/nips2023_shape_vs_texture/README.md`

**Computation**    Our modified texture synthesis program can be found under `https://github.com/Crazy-Jack/nips2023_shape_vs_texture/texture_synthesis_visualization/README.md`. We perform the optimization within half minutes for each image, especially with the usage of early stopping to prevent overfitting. Our computations were run on a NVIDIA A6000 machine.

## 6.2    Sparse CNNs on Shape Bias Benchmark

**Dataset**    We performed the experiments under the dataset provided by [4, 3]. In particular, it consists of 16 categories (airplane, bear, bicycle, bird, boat, bottle, car, cat, chair, clock, dog, elephant, keyboard) of cue-conflict images with each class consists 80 different images. Each image is style transferred with textures attributed to another categories. In order to correctly recognizing shape, the network needs to rely on the shape of an image rather than the texture information to make predictions. It pose additional challenge for the network as CNNs have the tendency to bias towards textures. The dataset is under CC BY 2.0 license.

**Network Architecture used for Inference**    In this section, we modifed VGG16 [9] and AlexNet [6] architecture during inference. We added sparsity Top-K operation layer inside the intermediate activation layers. The detailed architecture can be found at `https://github.com/Crazy-Jack/nips2023_shape_vs_texture/cnns-inference-top-k/network_architecture.txt`. Our model is reproducible by following the instructions in the github repo `https://github.com/Crazy-Jack/nips2023_shape_vs_texture/README.md`

**Computation**    We run the inference on a NVIDIA V100 machine. Each inference with / without Top-K takes about 10s to run. The experiment can be reproduced through running the code inside `TopkInference` folder of `https://github.com/Crazy-Jack/nips2023_shape_vs_texture`

## 6.3    Top-K bias training of the recognition networks

**Dataset**    In this experiment, we utilize two non overlapping subsets of ImageNet-1k [2]. For each class, we include all training and evaluation images. Here we documents the class names utilized for the experiments:

- IN-$S_1$:  n01440764, n02102040, n02979186, n03000684, n03028079, n03394916, n03417042, n03425413, n03445777, n03888257
- IN-$S_2$:  n01871265, n03063599, n03065424, n03100240, n03201208, n03208938, n03216828, n03249569, n03443371, n03447721

The data is available for free to researchers for non-commercial use.

**Network Design and Optimization Hyper-parameters**    We use conventional ResNet-18 as the backbone for the encoder but we change the first 7x7 2D convolution to 3x3 2D convolution and remove the first max pooling layer in the normal ResNet-18 (See code for details). This amplified the texture bias and shows a larger improvement of Top-K operation for the stylized image classification task, i.e. the shape bias test. Optimizations are handled via Stochastic Gradient Descent with a Cosine-Anneling learning rate scheduler starting from initial learning rate of 0.1. All experiments are run 3 times to produce the error bars. We install a 20% Top-K layer in the second layer of the modified ResNet-18 (early layer). Our model is reproducible via the github repo provided at `https://github.com/Crazy-Jack/nips2023_shape_vs_texture/README.md`

**Computation**   We run our experiments on NVIDIA A6000 machines. Each experiment took 3.5 hours to run 50 epochs. Each experiment is run 3 times to produced the error reported in the Table 1 in the main text. The experimental code is located under the `TopkRecognitionTrain` folder of the github repository.

### 6.4   Towards Shape Biased Few Shot Image Synthesis

**Dataset**   For better demonstrating the importance of the shape bias inside generator, we utilize images from ImageNet-1k [2] and MSCOCO [7]. A sample of the image used can be found in main text Figur 6(a). The full list of image name used are hosted at `https://github.com/Crazy-Jack/nips2023_shape_vs_texture/few-shot-img-syn/data/README.md`. ImageNet data is available for free to researchers for non-commercial use. MSCOCO is under Creative Commons Attribution 4.0 License which can be accessible at `https://creativecommons.org/licenses/by/4.0/legalcode`.

**Network Architecture and Optimization Hyper-parameters**   We utilize the FastGAN [8] architecture, with a modification of the generator architecture. In particular, we install the Top-K operation layer after the $32 \times 32$ resolution layer (`feat_32`, see code for more detail). Various Top-K hyper-parameters are adopted for different image sets as the best parameter for sparsity various dataset by dataset. For Jeep-100, Fish-100, Train-100, we utilize a sparsity of 5% whereas for the Table-100, the best sparsity resides in 15%. We further run ablation study for the influence of different sparsity to the image synthesis performance and document the results in Section 4. Our model is reproducible via the github repo provided at `https://github.com/Crazy-Jack/nips2023_shape_vs_texture/README.md`

**Computation**   We employ NVIDIA 4090 GPU for the computation. For 80,000 iteration steps, the program took about 7 hours to run. We observe 1 hour speed up by employing the sparsity Top-K operation. Further improvement was achieved by avoiding explicitly zero masks and multiplication with the activation. We took the generator of best synthesizing quality throughout the entire training period and report the FID and KID scores. Multiple run (3 times) are replicated to produce an error bar of the quantitative evaluation shown in main text Table 2.