# OpenReview forum: "Emergence of Shape Bias in Convolutional Neural Networks through Activation Sparsity"
_NeurIPS.cc/2023/Conference — NeurIPS 2023 oral_

### Official Review · Reviewer_oHBy · 2023-07-05

**Soundness:** 3 good
**Presentation:** 3 good
**Contribution:** 4 excellent
**Rating:** 7
**Confidence:** 3

**Summary:**

This work is motivated by the observation that deep neural networks may have a bias towards texture in classification decisions. The paper proposes a top-k sparsity selection process that is shown to result in emergence of shape sensitive neurons while improving performance across a variety of tasks.

**Strengths:**

1. The demonstration that shape encoding occurs when the proposed sparsification is applied even in the absence of training is an important result and contribution.
2. The paper demonstrates the value of this operation across a variety of tasks from recognition to synthesis and connections to neuroscience.
3. The paper is generally well presented and gives a variety of examples that present a strong case for the claims made.

**Weaknesses:**

1. There are some editorial changes that would benefit the paper (e.g. which numbers are bold in tables, when is SD bold etc.) and some places in the text where the description could be refined.
2. It would be nice to see something like figure 4 for the ResNet variants given their importance in the literature
3. Some references might be added (e.g. Md Amirul Islam, Matthew Kowal, Patrick Esser, Sen Jia, Björn Ommer, Konstantinos G. Derpanis, Neil D. B. Bruce, Shape or Texture: Understanding Discriminative Features in CNNs, ICLR, 2021.)

**Questions:**

1. How does the rank based selection for sparsity compare with 'softer' methods for inducing sparsity (e.g. L1 regularization)?

2. Is there any means of knowing the proper value for the k hyperparameter?

**Limitations:**

The authors are clear about limitations of the work.

---

> ### Author Rebuttal · Authors · 2023-08-10
>
> We appreciate the reviewer's kind comments and insightful feedbacks. We address the reviewers' concerns as follows:
>
> - ResNet variant of Figure 4. We thank the reviewer for bringing up this point. We conduct experiments on ResNet-50 (See general response Figure 1 (a)) and also on ViT (General response Figure 1 (b)).
>
> - L1 regularization: Indeed, we experimented with another soft version of sparsity, i.e. apply L1 norm to the loss function and document the results in the general response Table1 (5). We observe that the L1 induced sparsity introduces on-par shape bias as the Top-K do. We will perform more experiments on the effect of L1 activation loss on other tasks such as synthesis in the future.
>
> - Hyperparameter K. We currently perform grid search of the parameters but keep the Top-K value the same across different layers. We do observe that in ViT architecture, the fine shape bias would depend on the highest sparse level, e.g. if layer 1 keeps 20% of the activation and layer 2 keeps 40% activities, the final performance would be very similar to the one that both keep 20% of the activities. The underlying reason is still under investigation.
>
> - Editorial suggestion. We will further correct these minor errors and incorporate the suggested related work.

---

### Official Review · Reviewer_ZUQs · 2023-07-06

**Soundness:** 2 fair
**Presentation:** 3 good
**Contribution:** 3 good
**Rating:** 7
**Confidence:** 3

**Summary:**

The paper proposes to use activation sparsity induced by the top-k operation to address the recently identified hard problem in DNNs, the texture bias. The operation is simple yet highly effective, as it shows good structural encoding capability (Sec 4.1), shape bias (even without training, Sec 4.2) and parts/subparts learning dynamics (Sec 4.5). Finally the operation (with training) is validated with tasks including style robustness (ResNet-18, Sec 4.3) and few-shot image synthesis (FastGAN, Sec 4.4) on ImageNet subsets, where it also shows promising results.

**Strengths:**

+ [Originality] The paper is sufficiently novel in my opinion. Although the idea of top-k sparsity has been commonly used in various applications (training/inference acceleration, brain-inspired algorithms, etc.), the fact that the simple idea works for this new, challenging topic should be interesting to the community.
+ [Clarity] The paper is clear and easy to follow. The effort to clearly explain the motivations (e.g. Sec 4.4) and results (e.g. Sec 4.5) of the experiments is relatively noteworthy.
+ [Quality] The paper is overall of good quality in my opinion. The proposed method is well motivated, simple and effective. The evaluation, though purely empirical, is relatively comprehensive and promising.
+ [Significance] Given the simplicity and effectiveness of the proposed method, and the relevance of the topic, I think this work is significant and could be valuable to many in the community.

**Weaknesses:**

- [Evaluation, Minor] To further improve the impact of the paper, the authors could consider improving the following aspects.
1) Fig 2 and A.1 are both valuable yet limited. The authors could consider including larger-scale experiments/visualization and quantitative analysis of the results (e.g. textureness [38]).
2) The authors could consider including experiments combining with existing solutions (e.g. data augmentation based, given the orthogonality) on the benchmark [10, 39] to demonstrate this work’s potential to further boost performance, including on ViT-based architectures.

[38] The Synthesizability of Texture Examples, CVPR, 2014.\
[39] https://github.com/bethgelab/model-vs-human/

**Questions:**

1) Does the value of non-top-k units (now set to 0 in Eq 1) affect the experimental results (e.g. by overly shifting the mean or other statistics)? Shouldn’t some form of corrective offset (or learnable value other than 0) be applied?
2) How does the top-k without training performance compare to others in Table 1?
3) What do the colors mean in Fig 7?

**Limitations:**

The proposed method itself doesn’t seem to have serious limitations as far as I can tell. However, the limitations of the standard protocols & datasets [10] currently used to analyze the results could be discussed in the paper.

---

> ### Author Rebuttal · Authors · 2023-08-10
>
> We thank the reviewer for the kind advice and constructive feedback. The reviewer consider our paper is technically solid and has novel contribution as well as good writing. We prepare the following feedback to address the reviewer’s remaining questions:
>
> - Quantitative analysis of the visualization results. We agree with the reviewer however due to the page limits, we will put the further quantitative results in our supplementary.
>
> - Combining the existing solution. We appreciate the comment of the reviewer. Indeed, we generalize our method to more architecture (ResNet and ViT) in the general response. We will apply our methods to SIN in our anonymous github repository during the discussion period to further demonstrate that our method can be combined with existing solutions.
>
> - Non top-k value offset. Our motivation is to mimic the efficient coding principle utilized inside the brain. A learnable offset would introduce further optimization instability. Also during training, setting the non-top-k to zero would make the gradient to become zeros automatically. However, it might improve the inference accuracy if we keep the distribution mean statistics during Top-K inference. However, due to the limited time, we haven’t observed any significant difference in our preliminary experiments and we will do more tests in the following days.
>
> - TopK without training. We appreciate the reviewers’ point. Please refer to our general response Table 1 for more details. In general, we observe that our optimal parameters for Top-K training is not optimal for inference but training with Top-K significantly improves both shape bias and accuracy (as shown in (2) and (4) in the general response Table 1).
>
> - Colors in Figure 7. The white indicates positive and black indicates negative while the gray indicates 0.
> - The limitation of the standard protocols & datasets includes the fixed and limited category as well as its practically in the real world. We will include this limitation and propose more comprehensive in the future work.

---

### Official Review · Reviewer_muuS · 2023-07-06

**Soundness:** 4 excellent
**Presentation:** 4 excellent
**Contribution:** 3 good
**Rating:** 7
**Confidence:** 4

**Summary:**

Current deep learning models are known to have more texture bias for object recognition tasks while humans have more shape bias. They test the hypothesis that sparse coding may improve shape bias in deep learning models. Using Top-K operations, they show improved accuracy in pre-trained vison models as well as improved shape bias in models trained from scratch. Furthermore, in few-shot image systhesis tasks, models with sparsity constraints result in more structurally coherent images. The Top-K operations may be imposing binary masking on object parts, which can induce shape bias.



**Strengths:**

- Connecting sparsity with shape bias is novel.
- The authors examine both classification and generative models based on the sparsity operations.
- Their study has implicaitons in both building computational models and understanding human vision. Their empirical results suggest that sparsity can lead to improved shape bias, making the models potentially make judgments closer to humans. Conversely, their study suggests sparse encodings may play a role in inducing shape bias in human vision.
- Presentation is clear with helpful figures. I enjoyed reading it.



**Weaknesses:**

- Section 4.1: Additional comparisons with Ts w. Top-K only will be useful. It is unclear whether less structural information in TS w. Non Top-K is just due to a lack of representational power.
- Section 4.2: As noted in line 160, Vision Transformers show improved shape bias. How would you apply Top-K operations to Vision Transformers and do you expect to obtain more shape bias for those models? Since they are a popular model family, I think it's worth adding experiments or adding notes on them.
- Section 4.3: Does the Top-K training lead to many dead neurons, essentially reducing the network size, which would work as an implicit regularization?
- Section 4.5:  The idea suggested here seems related to models developed with topographic constraints (Margalit et al., 2023), where spatially smooth response patterns are learned. Comments on these models will be useful.
- Section 4.6: The discussion here implies that an ideal sparsity rate may be different for each layer. Applying different K values depending on layer depth or not applying sparsity to some layers in the experiments (e.g. Figure 3) will be interesting to test the hypothesis further.
- While I do think the study suggests an interesting perspective on shape bias using sparsity, the study may have less impact if models trained on larger datasets show more shape bias. For instance, in Figure 3, I wonder ConvNexts or ViTs trained on LAION-2B would give a stronger baseline.


Minor comments
- Line 53 ("indeed can indeed introduce"): indeed is repeated
- Line 161 ("already can already induced"): already is repeated
- Figure 4: I assumed it indicates humans, but the label for purple markers is missing.




**Questions:**

- Section 4.2: How does the number of correct recognitions change when sparsity is imposed on the models? Are we sacrificing accrucy a lot for better shape bias?
- Figure 3: For the boxplots, please explain the distribution, whether it is over multiple image categories or multiple seeds for applying sparsity, etc.


**Limitations:**

Currently, the study tested only convolutional architectures. Otherwise, limitations are adequately addressed.

---

> ### Author Rebuttal · Authors · 2023-08-10
>
> We thank the reviewer for the insightful comments. The reviewer think our paper is novel and solid. Here we prepare the response to address the reviewer’s questions.
> - Comparison with  Ts w. Top-K only. As for the limitation of the response pages, we put the further experiments under our anonymous github repository:
> https://anonymous.4open.science/r/nips2023_shape_vs_texture-1057
> - Apply the Top-K to the transformer architecture. We conducted experiments and observe similar improvements on the shape bias score when applying sparsity to ViT. Please refer to our general response on the details.
> - Dead neurons and implicit regularization. Yes, indeed. We also address this issue in our general response.
> - Margalit et al., 2023: We will certainly include this work as a related work in our camera ready version.
> - Different layer of sparsity. Yes, we indeed apply different Top-K to different layers. We intend to mimic the long range communication yet local dense computation of the brain, i.e. we divide the network to several chunks and apply sparsity at the connections of these chunks. In VGG for example,  we apply the Top-K layer after the 5th, 10th, 17th, 23th, 31th original layers (after each MaxPooling Layer). In AlexNet, we also apply the Top-K layer after the MaxPooling except adding 2 additional Top-K layers at the 8th and 10th layer to make the sparse layer more evenly distributed. For the synthesis network, we only apply the sparsity on the 4th layer out of 8 layers as we observe that applying severe sparsity near the output layer would yield suboptimal performance.
> - Stronger baseline. We agreed on a strong baseline for evaluation. We included the ViT results in the general response. We believe our method is orthogonal to the scaling approach and our further study shows it can be combined with larger models to further improve the shape bias. However, due to the time limitation, we were unable to experiment with other large models that are trained on LION2B for example. We will try our best to apply the proposed method on the extremely large scale trained models in the following discussion period.
> - Accuracy v.s. Shape Bias. Our main manuscript Section 4.3 indicates that a network that is trained with Top-K operation can achieve competitive accuracy while increasing the shape bias robustness. In the general response Table 1, we can observe that Top-K training can lead to significant improvement compared to Top-K inference (see General Response Table 1 (2) v.s. (4)).
> In terms of accuracy and shape bias trade-off, we agree with the reviewer that applying sparsity directly to inference could lead to degraded classification results. We interpolate between no sparsity and severe sparsity in the general response Figure 2(b). We vary the number of sparsity layers from 0 to 5 and observe that there is a sweet spot for balancing the shape bias and the classification accuracy. Therefore, in practice, we recommend training or fine tuning with Top-K operation to achieve the best performance and shape bias simultaneously.
> - The boxplot in Figure 3 and typos. We appreciate the reviewer’s question. To be specific, the distribution of the box plot is over different image categories. We will make further clarification and also correct the typo in our camera ready version.

---

### Official Review · Reviewer_24RM · 2023-07-11

**Soundness:** 3 good
**Presentation:** 4 excellent
**Contribution:** 4 excellent
**Rating:** 8
**Confidence:** 3

**Summary:**

This paper proposes enforcing sparsity within a neural network through a top-K mechanism and demonstrates that this leads in an emergent fashion an emphasis on shape rather texture in the learned representations of the network. This in turn leads to improvements in robustness for network behaviour.

**Strengths:**

- The paper is clearly written and laid out.

- The paper presents a novel and innovative contribution to a challenging goal in deep learning (namely, shifting neural networks more toward shape representations rather than texture representations). Although this topic has been studied before, as the paper points out existing methods to achieve this have significant drawbacks (such as the practicality of style-transfer augmentations), whereas the method proposed in this paper is extremely practical to implement.

- The use of multiple different styles of experimental methods helps round out the conclusions (for example, Section 4.1 is a relatively straightforward qualitative experiment that is nevertheless compelling, naturally leading to the more comprehensive later experiments.

-

**Weaknesses:**

- I found it wasn't always clear whether Top-K was being applied to all layers or a subset of layers. Section 4.4 explicitly mentions that Top-K was used only in the fourth layer; why this layer, and how was this determined?

**Questions:**

- In Figure 6(b) is there a difference between Ours 1 and Ours 2, or is that simply two different examples of output?

- Does the use of a top-K mechanism lead to some of the network essentially becoming redundant (i.e. always ignored) and wasted computation?

**Limitations:**

The paper does not have any obvious limitations that were not discussed.

---

> ### Author Rebuttal · Authors · 2023-08-10
>
> We thank the reviewer for the positive review and constructive feedback. The reviewer thinks our paper is technically solid with excellent contribution to a challenging goal in deep learning. We carefully prepare the following response aiming to address the reviewer’s questions:
>
> - Top-K layer specification:
>
>    We apply the Sparsity layer in a subset of the network. It is based on the intuition that the brain utilizes sparsity for long range communication but can allow local dense computation. We divide the networks into chunks where within each chunk the neuron’s activities are allowed to be dense (keep original) but the communication across different chunks is set to be sparse. To be specific, in the VGG Network, we divides the network into 5 chucks and apply the Top-K layer after each MaxPooling layer. Specifically, across a total of 31 feature layers in VGG, we apply the Top-K layer after the 5th, 10th, 17th, 23th, 31th original layers. In AlexNet, we also apply the Top-K layer after the MaxPooling except adding 2 additional Top-K layers at the 8th and 10th layer to make the sparse layer more evenly distributed. In the ResNet-18 training, we only apply the Top-K layer after the second layer to demonstrate the concept of sparse specialist code. We also apply the Top-K layer only in the 4th layer of the GAN generator as we observe empirically that it induces the best performance across other settings (such as applying multiple Top-K layers or the less severe sparsity).
>
>     Interestingly, our new experiments as shown in the Figure 1 in the general response indicates that ViT architectures seems to requires more dense activation, as we can observe that a moderate to high density (keep 50% - 80% of the original ranking neurons) yield the best performance for most of the classes (best mean shape score is achieved by keeping 60% of the original activation).
>
> - Difference in  Figure 6(b): We apologize for the confusion. Ours 1 and Ours 2 are from the same setting. We add more specification and synthesis examples in the camera ready version.
>
> - Redundancy in the network (Please see the general response).

---

### Author Rebuttal · Authors · 2023-08-10

We appreciate the reviewers’ kind responses and constructive comments. In the following, we provide further experiments to support our paper.

First, we generalize section 4.2 to ResNet-50 and ViT-B architectures. The results are documented in Figure 1 of the general response. The ResNet-50’s sparsity definition is the same as AlexNet and VGG. For ViT-B, we reshape the activation response from [n, h * w, d] to [n, d, h * w]  and apply the Top-K selection over dimension 2 before the activation is passed through the multiple head attention (Note that h and w is the height and weight of the latent tensor after reshape it to 2d, for ViT-B with patch size 16 on the 224x224 images, h=w=14). Please refer to our updated code for precise implementation. We observe that the similar shape bias gain from applying Top-K operation to ResNet-50 and ViT-B although ViT-B needs denser activation compared to the ConvNets.

We further add experiments for 4.3. Specifically, we add experimental results (2), (3) and (5) as highlighted in blue color. We first test how the model behaves when we apply Top-K operation on Non-Top-K trained models as Reviewer ZUQs points out. As the choice of hyper-parameter K in the proposed Top-K operation is critical to the model's performance, we apply the same Top-K parameter setting for (2) and (4). It can be observed that inference with Top-K with sub-optimal setting (2) would sometimes induce performance deficit but the models' performance will improve when applying the same Top-K sparsity during training (3). Second, we test the hypothesis that whether other soft sparsity constraints like L1 regularization would have the same effect (as suggested by Reviewer oHBy). We introduce a L1 term $ + \lambda || \texttt{NeuroActivation} ||_{1}$ during training (we tried $\lambda$ choice of $\{10, 1, 0.1\}$ and found $\lambda=1$ works the best.) and provide its results in (5). We can observe that (5) has similar performance as trained with Top-K layers (4), suggesting that L1 regularization induced sparsity could also help for improve the out-of-domain generalization of the model.

Next, we investigate the internal activation when applying Top-K during training. In General Response Figure 2(a), we find that the Top-K training would lead to “dead neurons”, i.e. the neurons that will not be used across 100k inputs. We produce this plot by logging the frequency of the times each neuron is used. Specifically, we take the GAN generator trained in Section 4.4 and pass 100k random noise from the standard normal distribution. As the network is trained to map the noise to a real image (jeep class), we analyze the activation tensor inside the Top-K layer. We consider each neuron to be used once if it’s inside the Top-K selection. We normalize this utilization number by 100k and get the utilization frequency for each neuron. We can see that 15% of neurons are alive and the rest doesn’t respond at all. Therefore we can understand our proposed method as an implicit regularization to reduce the model size.

Finally, we also provide analysis between the trade-off between shape bias gain and the classification accuracy sacrifice (General Response Figure 2(b)). We vary the number layers we install Top-K during inference on the AlexNet and evaluate the mean shape score as Figure3 in the main text and the normal classification accuracy. We can observe that there is a sweet spot that can balance the shape bias score and the classification accuracy. Combining with the experiments from general response Table1 (2) and (4), where Top-K training could increase both classification accuracy and the shape bias score, we recommend training or fine tuning with Top-K to obtain improvement on both classification accuracy and the shape bias robustness.

---

### Decision · Program_Chairs · 2023-09-21

**Decision:**

Accept (oral)

**Comment:**

The paper shows that enforcing activation sparsity in deep neural networks biases their decision-making towards using shape cues, similar to human perception. All reviewers agree that it is a strong paper that makes a valuable contribution. They noted some minor concerns which were addressed well during the response and discussion period.